# Predictive Values of Nocturia and Its Voiding Frequency on the Aging Males’ Symptoms

**DOI:** 10.3390/ijerph191811632

**Published:** 2022-09-15

**Authors:** John Wai-Man Yuen, Ivy Yuen-Ping Wong, Peter Ka-Fung Chiu, Jeremy Yuen-Chun Teoh, Chi-Kwok Chan, Chi-Hang Yee, Chi-Fai Ng

**Affiliations:** 1School of Nursing, The Hong Kong Polytechnic University, Hung Hom, Kowloon, Hong Kong; 2S.H. Ho Urology Centre, Department of Surgery, The Chinese University of Hong Kong, Shatin, Hong Kong

**Keywords:** nocturia, aging male symptoms, quality of life, health-related quality of life, male adults, NQoL, urinary frequency, bedtime urination, sleep

## Abstract

Background: The link between nocturia and aging male symptoms (AMS) has not been scientifically established. This study aimed to measure the degree of severity of AMS that impacts health-related quality of life (HRQoL) in adult males living with nocturia and to determine the predictive values of nocturnal factors on AMS. Methods: This is an extended analysis of new data collected by using the Hong Kong Traditional AMS (HK-AMS) scale and the Cantonese version of the Pittsburgh Sleep Quality Index (PSQI) in a recently published cross-sectional population-based survey. Results: Of the 781 respondents that completed the set of questionnaires, 68% and 61% of men living with nocturia reported clinically significant (at moderate-to-severe levels) somato-vegetative and sexual AMS; the prevalence and severity were increased with advancing nighttime voiding frequency. Age, the Global PSQI score, certain metabolic diseases, the nocturia-specific QoL (NQoL) score and bedtime voiding frequency were found to be significant predictive factors for composite somato-vegetative and sexual AMS. Conclusions: The current findings suggested the inclusion of nocturia when measuring male-specific HRQoL related to aging.

## 1. Introduction

Nocturia causes many public health concerns because of its impact on quality of life (QoL), in addition to its associations with numerous illnesses and conditions [1]. Nocturia is often co-morbid with sleep disturbance, significantly impacting life activities and functional levels [2]. Moreover, accumulating evidence links nocturia with certain health risks. Nocturia is a known risk factor for falls, particularly in older people [3]. However, recent research conducted in Korea extended the risk associations of nocturia with slipping and falling to all ages of adult males [4]. In this relation, the Nocturia Quality of life module of the International Consultation on Incontinence Modular Questionnaire (ICIQ-NQoL) has shown to be a useful tool for predicting the risk of falling [5]. Owing to its implications on sleep, nocturia was identified as a stronger predictive symptom than snoring for obstructive sleep apnea. Bedtime voiding frequency was even found to be reflective in the severity of broad sleep-disordered breathing [6,7]. Sleep disturbance caused by nocturia negatively impacts one’s overall well-being, general health, vitality, and essential biological rhythms [8,9].

Our recent research reported that 63% and 80% of Hong Kong adult males, respectively, were living with nocturia [10] and aging male symptoms (AMS) [11] and that both were correlated with age; however, both also shared strong associations with several metabolic and urological conditions which harmed the health-related quality of life (HRQoL). All these conditions were linked with a decline in testosterone levels in men [12]. Testosterone levels decline with age, resulting in the deterioration of men’s health [13,14]. Accumulating evidence supports the benefits of testosterone replacement therapy for treating symptoms of late-onset hypogonadism and improving lower urinary tract symptoms (LUTS) [15] and numerous metabolic conditions, such as insulin resistance, adiposity, and dyslipidemia [16]. The total serum testosterone level was found to be negatively correlated with prostate volume, suggesting a relationship with benign prostate hyperplasia (BPH), insulin level, and an array of obesity-related factors that further linked testosterone to metabolic syndrome [16,17,18]. On the other hand, testosterone also interplayed between nocturia and sleep. The production of testosterone was interfered with when sleep was fragmented to a level failing to show rapid eye movement (REM), as caused by nocturia [19]. Interestingly, recent research indicated the dual roles of testosterone on sleep quality that could be affected by both deprived and excessive supplies in the circulation, whereas obesity was shown to exhibit mediating roles in the inter-correlation between serum testosterone and sleep efficiency [20]. Well-designed sleep restriction experiments clearly show the effects of sleep deprivation on testosterone reduction [21], which may also lead to other health impacts associated with nocturia and AMS. The ICIQ-NQoL questionnaire consisted of two subscales, with one particular measure of the sleep/energy factor [22]. There were also two items in the ‘Aging male’ symptoms (AMS) scale specifically asking about the sleep problem and its consequence [23]. Within the context of men’s HRQoL, considering testosterone may play a central role in nocturia to form part of the AMS. The objective of this study was to determine the predictive values of nocturia and its related factors on AMS.

## 2. Materials and Methods

### 2.1. Study Design

This is an extended study of the community-based survey on nocturia that our research team conducted recently [10]; it involves the analysis and interpretation of unpublished data set on AMS and sleep quality that was collected among adult males who were reported as suffering from nocturia. In accordance with the street-intercept and random walk design, respondents with nocturia were also invited to complete the AMS scale and the Pittsburgh Sleep Quality Index (PSQI) on-site after administering the NQoL questionnaire and collecting the demographic information.

### 2.2. The Instruments and Measurements

The Cantonese version of ICIQ-NQoL questionnaire was used to measure frequency of nocturia and NQoL overall and factor (factor 1: sleep/energy; factor 2: bother/concern) scores [10]. Following the original ICIQ-NQoL construct, frequency of nocturia was counted as 0, 1, 2, 3, or ≥4 episodes of urination per bedtime, which is considered a categorical variable during the data analysis.

The Hong Kong Traditional AMS (HK-AMS) scale [11] and the Cantonese version of the PSQI (CPSQI) [24] were adopted in this study.

The HK-AMS scale consisted of 17 items using the 5-point Likert scale (from 1 to 5) of ‘severity’ to measure the personal perception of respondents on male symptoms or complaints associated with aging. The scores of relevant items were summed to generate the composite score (range of 17–85) and 3 domain scores in 3 dimensions: somato-vegetative (range of 7–35), psychological (range of 5–25), and sexual (range of 5–25). Higher score represented a higher severity of the aging symptoms, whereas the composite score can be further categorized into 4 severity levels as ‘no significant symptoms’ (<26), ‘mild’ (27–36), ‘moderate’ (37–49), and ‘severe’ (>50). Furthermore, the domain scores were categorized into different severity levels according to Heinemann et al. [23]. The psychometric prosperities of similar male population were reported by Yuen et al. [11].

The 19-item CPSQI is a well-validated questionnaire that has been used for measuring the subjective sleep quality of different Chinese populations [24]. The items were categorized into 7 component scores (each ranging 0–3), which are summed to produce a 0–21 global score range, with any scores >5 indicating poor sleeper.

### 2.3. Data Processing and Statistical Analysis

Data collected from the survey were entered and analyzed using SPSS version 25.0 (IBM, Armonk, NY, USA) and Prism version 9.0 (GraphPad, San Diego, CA, USA). Descriptive statistics were used for reporting the categorical variables (frequency and percentage) and continuous variables (mean and standard deviation (SD)) of the demographics, bedtime voiding frequency, NQoL scores, AMS, and sleep conditions. Significant differences among nominal and continuous variables between groups were evaluated by using Chi-squared (χ^2^), Student’s *t*-test, and one-way ANOVA accordingly. Pearson’s correlation analysis was used to assess the linear correlations among continuous variables. Furthermore, stepwise multiple regression analysis was performed to determine the multicollinearity among different variables to identify the significant predictive factors for AMS in the studied population.

## 3. Results

Those among the studied population that reported prevalence rates of 1, 2, 3, and ≥4 voiding episodes per night were 50.4%, 32.5%, 12.3%, and 4.7%, respectively [10]. Among the 1239 adult men who participated in the survey, all had completed the AMS scale, while the CPSQI was completed by 65.5% and 75% of those with and without nocturia, respectively. Table 1 summarizes and compares the demographics, health conditions, AMS, and sleep quality between non-nocturia and nocturia respondents. The results demonstrated the strong association between nocturia and a spectrum of urological disorders; benign prostatic hyperplasia (BPH) was the most frequently reported by men with nocturia (Table 1). When compared with the non-nocturia respondents, nocturia was also shown to be statistically significantly (*p* < 0.001) associated with scores and severity levels of all AMS and sleep quality measures (Table 1; Appendix A).

### 3.1. High Prevalence of Somato-Vegetative and Sexual AMS among Adult Males Living with Nocturia

The studied male population was not only living with nocturia but also had a mild level of total AMS, with a mean composite score of 31.6 ± 8.4; moderate level of somato-vegetative symptoms, with a mean score of 14.9 ± 4.3; and moderate-to-severe level of sexual symptoms, with a mean score of 9.2 ± 3.6 (Table 1). In addition, almost half of the nocturia population was rated as poor sleepers, with a mean PSQI score of 6.7 ± 3.3 (Table 1).

In comparison, AMS was found to be more commonly *(p* < 0.001) reported by the respondents with nocturia than those without, particularly with 25% at the moderate level (versus 7% of non-nocturia) and 3% at the severe level (versus 0.4% of non-nocturia) (Appendix A). The prevalence rates of moderate-to-severe levels of somato-vegetative, sexual, and psychological symptoms were measured as 68%, 61%, and 29%, respectively. As shown in Figure 1, the composite AMS, somato-vegetative, sexual, and Global PSQI scores shared similar significant upward trends (*p* < 0.001) with increasing bedtime voiding frequency, while all of them except the psychological domain were significantly (<0.001) varied among the episodes of bedtime urination. The linear increase in composite AMS scores plateaued at three voiding episodes per night (Figure 1a). Over 80% of men having 3 voids per night were rated with mild-to-moderate levels of AMS (about half and half in ratio), while the highest percentage of severe AMS was dominant among those having ≥4 voids (Table 2). Although a clear increasing trend was observed with the advancing bedtime voiding frequency for the somato-vegetative symptom scores (Figure 1b), the percentage of the clinically significant (i.e., moderate-to-severe) symptom level was increased from 60% to 76–80% discretely between those with a 1 and ≥2 voiding frequency (Table 2). Psychological symptoms were the least affected by the bedtime voiding frequency. A slight, gradually increasing trend was observed for both the domain score (Figure 1c) and severity classification (Table 2). The prevalence of moderate-to-severe level sexual AMS peaked at 84.3% (51% were at severe level) among those with a voiding frequency of 3 (Table 2). The sexual symptom score followed the trend observed in the Global PSQI score that was linearly increased at voiding frequency ≤3 and significantly dropped (*p* < 0.001) at frequency ≥4 (Figure 1d,e). The same trend was followed by the prevalence of poor sleepers who had a Global PSQI score >5 (Figure 1f). Particularly, almost 10% of men with nocturia used medications to help their sleep, which was significantly (*p* < 0.001) more common than those without nocturia at 2.6% (Table 1). Furthermore, the severity levels of the 7 PSQI components in accordance with the bedtime voiding frequency are provided in Appendix A.

### 3.2. NQoL Score and Voiding Frequency Are Predictors for Composite, Somato-Vegetative, and Sexual AMS in Addition to Age and PSQI Score

A correlational analysis indicated that the composite AMS and domain scores were moderately inter-correlated with the bedtime voiding frequency (r = 0.168–0.390; *p* < 0.001), the NQoL overall score (r = 0.228–0.424; *p* < 0.001) as well as its factor scores (r = 0.190–0.411; *p* < 0.001), and the Global PSQI score (r = 0.248–0.408; *p* < 0.001) (Table 3). Stepwise multiple regression was performed separately with the composite AMS and three domains as dependent variables. The independent variables were evaluated in the order of demographics and illnesses as ‘block 1’, bedtime voiding frequency and NQoL score as ‘block 2’, and Global PSQI score as ‘block 3’. Multicollinearity was observed between the overall NQoL score and its two factor scores (with r = 0.928–0.953; *p* < 0.001); therefore, only the NQoL score was evaluated by the stepwise multiple regression.

As summarized in Table 4, based on the standardized coefficient (β) values, from strongest to weakness, the significant independent predictive factors identified for: (1) composite AMS were the NqoL score, Global PSQI score, bedtime voiding frequency, HTN, smoking habit, DM, and age; (2) somato-vegetative AMS were the Global PSQI score, NQoL score, bedtime voiding frequency, comorbidity of DM and HTN, smoking habit, HTN, age, and BPH; (3) psychological AMS were the NQoL score, Global PSQI score, DM, and age; and (4) sexual AMS were the bedtime voiding frequency, Global PSQI score, HTN, NQoL score, and age. Two demographic factors (age and smoking habits) and two metabolic illnesses (diabetes mellitus and high blood pressure) were identified as significant predictive factors for various AMS domains when examined in block 1. However, in the final regression models, independent variables entered in block 2 and block 3 were found to have stronger predictive values than those tested in block 1, where the predictive values of demographics and illnesses became weaker and less significant at *p* > 0.05 (Table 4). HTN was only predictive for the somato-vegetative and became the least significant predictor of the presence of nocturia and poor sleep quality (Table 4). The current models identified the NQoL score and Global PSQI score as the two strongest predictors for the composite, somato-vegetative, and psychological AMS (Table 4). In relation to the NQoL score, bedtime voiding frequency was also shown to be a significant predictor important for composite, somato-vegetative, and sexual AMS (Table 4).

## 4. Discussion

This study found that many of the Hong Kong adult males living with nocturia were also co-morbid with moderate-to-severe levels of somato-vegetative and sexual AMS, which negatively impacted their HRQoL. Both NQoL and AMS questionnaires measured sleep quality, which was believed to be interplayed between nocturia and AMS, among other factors such as age and certain age-related illnesses. The NQoL factors, AMS severity, and sleep quality were shown to be strongly inter-correlated with each other. The current study identified, for the first time, that nocturnal factors (including NQoL scores, voiding frequency, and sleep quality) are strong predictors of AMS. Other well-known predictive risk factors for AMS were demonstrated to become weaker and less significant when those nocturnal factors were introduced into the regression models.

Clinically significant (moderate-to-severe) levels of somato-vegetative (68%) and sexual (61%) AMS were more prevalent in the nocturia population, as compared to more than 80% of the general adult male population living in the city with AMS at little-to-mild levels [11]. Nocturia commonly coexisted with age-related illnesses; in particular, diabetes and hypertension were both characterized by increased urinary frequency and sleep fragmentation [25,26]. Predictably, aging and age-related illnesses linked nocturia to AMS, which are both known to be associated with a decline in androgen [12,13,14]. The consequences of testosterone deficiency in causing nocturia and sleep disturbance were extensively reviewed by Shigehara et al. [27]. Nocturia was shown to be the only significant item of the International Prostate Symptom Score (IPSS) questionnaire that negatively correlated with the serum total testosterone in middle-aged men [28]. Pathologically, the increase in night voiding frequency in aging men could also be due to the interference of diuresis and the anatomical alteration of the lower urinary tract resulting from reduced androgen production, which in turn increases urine production during the night [29]. In this relation, the ‘aging males’ symptoms (AMS) scale was established to measure the symptoms of aging men and their impact on health-related Quality of Life (HRQoL), based on the assumption that men would develop specific complaints during the aging process due to androgen deficiency [23]. Around 10% of men with nocturia reported using medications to help their sleep (Table 1). Considering its influence on circadian rhythms, which is a known factor for lower urinary symptoms (LUS), a recent randomized, controlled trial suggested the therapeutic effects of melatonin on nocturia [30]. This effectiveness could be due to the increase in bladder capacity and decrease in urine volume, as demonstrated in rats treated with melatonin [31].

The high prevalence of nocturia and its correlation with age [10] explained, at least partly, the high prevalence of AMS; however, the strong inter-correlational relationships identified between the variables measured by the three questionnaires in this study further supported the cause-and-effect relationship that nocturia disturbs sleep to cause long-term consequences which harm physical and psychological health, hence affecting the overall HRQoL [32,33,34]. Particularly, results herein indicated that the AMS sexual and PSQI scores were both lower in people with nocturia ≥4, which could be explained by an older subpopulation (average age of 73 [10]) who sleep for shorter durations at night and are commonly less sexually active. The strong correlations of PSQI with both ICIQ-NQoL and AMS scales, as well as its independent predictive value on AMS, suggested sleep quality was important for the HRQoL of men. They also supported the assumption that the overall AMS-associated HRQoL could be modulated by sleep quality affected by nocturia. The ICIQ-NQoL alone measures the nocturia-specific QoL with a focus on two domains, namely, bothersomeness and sleep disturbance [22]. However, the AMS further assesses the general somato-vegetative, psychological, and sexual well-being of men [23]. The AMS scale consists of two items related to sleep disturbance, but none of the items asked about nocturia or related conditions. In the current study, the strong predictive values of NQoL, bedtime voiding frequency, and sleep quality on AMS suggested the combined use of the ICIQ-NQoL questionnaire and AMS scale for measuring the long-term impact on the HRQoL of adult males. Otherwise, items could be added to the AMS scale to cover the bedtime voiding frequency and its bothersome level. The ICIQ-NQoL has been proposed to be a useful tool for predicting the risk of falls [5]; therefore, the current findings support that the same can be used for AMS.

The current study reports, for the first time, the association between nocturia and AMS, which also identifies bedtime voiding frequency as a significant predictor for AMS. However, biological markers should also be evaluated to establish the interrelationship between nocturia and AMS. For instance, nocturia should be precisely measured by using the frequency volume chart (FVC) over a 2–3-day period [35]. Moreover, serum testosterone [36] and melatonin [37] levels should also be included in the predictive model to represent the modulating effects of androgen deficiency and sleep disruption, respectively. More in-depth investigations using cohort studies and interventional designs for further understanding of the underlying pathology on how nocturia and AMS are related to aging are warranted.

## 5. Conclusions

Nocturia was demonstrated to be associated with clinically significant somato-vegetative and sexual AMS, which might be explained by the androgen deficiency that is implicated with aging and sleep quality. In particular, the NQoL score and nighttime voiding frequency impacted the HRQoL of adult males, which were shown to be strong predictive factors for AMS. The current findings suggested that the occurrence of nocturia can be a reference indicator for poor male-specific HRQoL related to aging, especially when testosterone screening is unavailable.

## Figures and Tables

**Figure 1 ijerph-19-11632-f001:**
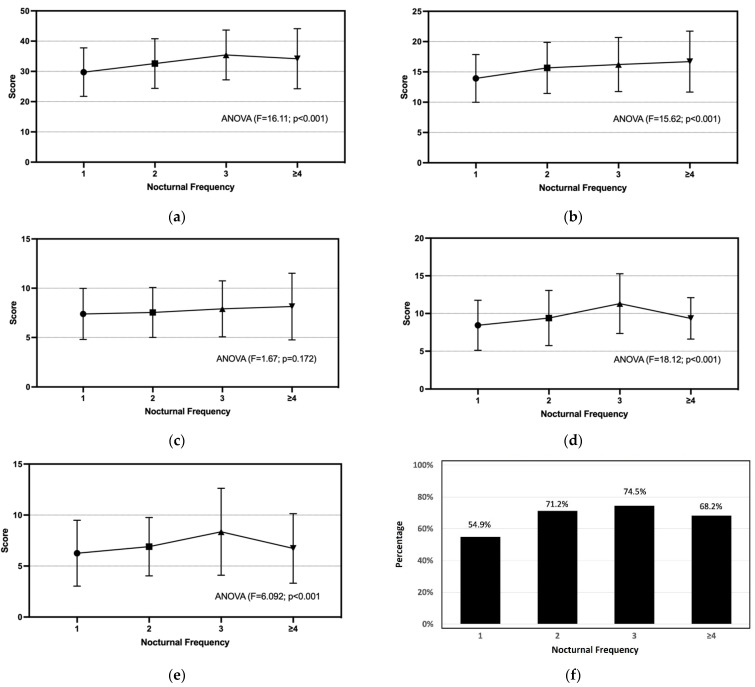
Severity of AMS and quality of sleep rated by the respondents with nocturia are presented as the scores of: (**a**) composite AMS, (**b**) somato-vegetative AMS, (**c**) psychological AMS, (**d**) sexual AMS, (**e**) Global PSQI, and (**f**) prevalence of poor sleepers against the increasing bedtime voiding frequency.

**Table 1 ijerph-19-11632-t001:** A summary of demographic, health-related characteristics, AMS, and sleep quality measures of the studied adult male population living with and without nocturia.

Variables	Total	Non-Nocturia	Nocturia	χ^2^
	*N = 1239*	*N = 458*	*N = 781*	*p*-Value
	Number (Percentage)			
Demographics and health condition				
Mean age ± SD, years	56.57 ± 15.22	49.83 ± 14.84	60.52 ± 14.01	<0.001
Married/cohabited	822 (66.3)	306 (66.8)	516 (66.1)	0.789
Education < college	750 (60.5)	217 (47.4)	533 (68.2)	<0.001
Current smoker	610 (49.2)	202 (44.1)	452 (57.9)	<0.001
Daily drinker	312 (25.2)	107 (23.4)	205 (26.2)	<0.001
Diabetes mellitus	181 (14.6)	34 (7.4)	147 (18.8)	<0.001
Hypertension	448 (36.2)	98 (21.4)	350 (44.8)	<0.001
Diabetes mellitus + hypertension	153 (12.3)	30 (6.6)	123 (15.7)	<0.001
Urological disorders ^#^	209 (16.9)	21 (4.6)	188 (24.1)	<0.001
AMS measures				
HK-AMS scale completed, n (%)	1239 (100)	458 (100)	781 (100)	N.S. ^†^
Mean composite score ± SD	29.29 ± 8.44	25.35 ± 6.85	31.59 ± 8.44	<0.001 ^‡^
Mean somato-vegetative score ± SD	13.71 ± 4.31	11.68 ± 3.56	14.90 ± 4.27	<0.001 ^‡^
Mean psychological score ± SD	7.22 ± 2.50	6.69 ± 2.13	7.54 ±2.65	<0.001 ^‡^
Mean sexual score ± SD	8.35 ± 3.44	6.98 ± 2.63	9.15 ± 3.60	<0.001 ^‡^
PSQI measures				
CPSQI completed, n (%)	856 (69.1)	344 (75.1)	511 (65.4)	<0.001
Global PSQI score	5.88 ± 3.06	4.71 ± 2.21	6.67 ± 3.29	<0.001 ^‡^
Poor sleeper (Global PSQI score >5)	575 (67.3)	186 (54.1)	389 (76.1)	<0.001
Sleeping medication (past month)	59 (6.9)	9 (2.6)	50 (9.8)	<0.001

^#^ Prostate cancer, BPH, renal stone, UTI, testicular problems; ^‡^ Student’s t-test; ^†^ non-significance.

**Table 2 ijerph-19-11632-t002:** Prevalence of different severity levels of composite AMS and its domains associated with bedtime voiding frequency.

AMS Domains	Bedtime Voiding Frequency				χ^2^
	1 Time	2 Times	3 Times	≥4 Times	*p*-Value
Number (Percentage)					
Composite					
No/Little	170 (43.2)	71 (28.0)	12 (12.5)	8 (21.6)	<0.001
Mild	151 (38.3)	102 (40.1)	37 (38.5)	18 (48.7)	
Moderate	67 (17.0)	76 (29.9)	43 (44.8)	6 (16.2)	
Severe	6 (1.5)	5 (2.0)	4 (4.2)	5 (13.5)	
Somato-vegetative					
No/Little	20 (5.1)	11 (43.3)	5 (5.2)	3 (8.1)	<0.001
Mild	143 (36.3)	51 (21.1)	14 (14.6)	5 (13.5)	
Moderate	171 (43.4)	124 (48.8)	45 (46.9)	18 (48.7)	
Severe	60 (15.2)	68 (26.8)	32 (33.3)	11 (29.7)	
Psychological					
No/Little	116 (29.4)	47 (18.5)	20 (20.8)	10 (27.0)	0.002
Mild	166 (42.1)	139 (54.7)	44 (45.8)	14 (37.9)	
Moderate	94 (23.9)	58 (22.8)	23 (24.0)	7 (18.9)	
Severe	18 (4.6)	10 (4.0)	9 (9.4)	6 (16.2)	
Sexual					
No/Little	69 (17.5)	30 (11.8)	6 (6.3)	5 (13.5)	<0.001
Mild	117 (29.7)	64 (25.2)	9 (9.4)	3 (8.1)	
Moderate	130 (33.0)	76 (29.9)	32 (33.3)	18 (48.7)	
Severe	78 (19.8)	84 (33.1)	49 (51.0)	11 (29.7)	

**Table 3 ijerph-19-11632-t003:** Correlation matrix for the bedtime voiding frequency, PSQI, ICIQ-NqoL, and AMS variables amongst respondents with nocturia.

Variables	Bedtime Voiding Frequency	Global PSQI Score	NQoL Overall Score	NQoL Factor 1 Score	NQoL Factor 2 Score	AMS Composite Score	AMS Somato-Vegetative Score	AMS Psychological Score
AMS sexual subscale score	r = 0.345; *p* ≤ 0.001	r = 0.248; *p* < 0.001	r = 0.228; *p* < 0.001	r = 0.190; *p* < 0.001	r = 0.212; *p* < 0.001	r = 0.775; *p* < 0.001	r = 0.476; *p* < 0.001	r = 0.417; *p* < 0.001
AMS psychological score	r = 0.168; *p* ≤ 0.001	r = 0.325; *p* < 0.001	r = 0.369; *p* < 0.001	r = 0.346; *p* < 0.001	r = 0.362; *p* < 0.001	r = 0.780; *p* < 0.001	r = 0.614; *p* < 0.001	
AMS somato-vegetative score	r = 0.391; *p* ≤ 0.001	r = 0.421; *p* < 0.001	r = 0.416; *p* < 0.001	r = 0.389; *p* < 0.001	r = 0.410; *p* < 0.001	r = 0.887; *p* < 0.001		
AMS composite score	r = 0.390; *p* ≤ 0.001	r = 0.408; *p* < 0.001	r = 0.424; *p* < 0.001	r = 0.386; *p* < 0.001	r = 0.411; *p* < 0.001			
NQoL factor 2 score	r = 0.253; *p* ≤ 0.001	r = 0.263; *p* < 0.001	r = 0.953; *p* < 0.001	r = 0.797; *p* < 0.001				
NQoL factor 1 score	r = 0.302; *p* < 0.001	r = 0.288; *p* < 0.001	r = 0.928; *p* < 0.001					
NQoL overall score	r = 0.295; *p* ≤ 0.001	r = 0.294; *p* < 0.001						
Global PSQI score	R = 0.320; *p* < 0.001							

**Table 4 ijerph-19-11632-t004:** Predictors of composite AMS and its domains identified by stepwise multiple regression.

AMS Domains (Final Model)	Predictors	b	SE	β	*p*-Value	Adjusted *R*^2^	Regression Significance
Composite	(Constant)	16.231	1.483		<0.001	0.335	<0.001
	NQoL score	0.252	0.036	0.273	<0.001		
	Global PSQI score	0.665	0.109	0.241	<0.001		
	Bedtime voiding frequency	1.267	0.338	0.161	<0.001		
	Hypertension (HTN)	1.427	0.757	0.081	0.060		
	Smoking habit	0.810	0.385	0.078	0.036		
	Diabetes mellitus (DM)	1.749	0.961	0.073	0.069		
	Age	0.036	0.023	0.065	0.120		
Somato-vegetative	(Constant)	7.354	0.758		<0.001	0.336	<0.001
	Global PSQI score	0.364	0.056	0.258	<0.001		
	NQoL score	0.120	0.019	0.253	<0.001		
	Bedtime voiding frequency	0.694	0.175	0.172	<0.001		
	Comorbidity (HTN + DM)	1.236	0.555	0.094	0.026		
	Smoking habit	0.488	0.197	0.092	0.014		
	Hypertension	0.478	0.405	0.053	0.238		
	Age	0.010	0.012	0.036	0.392		
	Benign prostatic hyperplasia	0.431	0.619	0.027	0.486		
Psychological	(Constant)	4.584	0.422		<0.001	0.188	<0.001
	NQoL score	0.080	0.012	0.291	<0.001		
	Global PSQI score	0.182	0.035	0.222	<0.001		
	Diabetes mellitus	0.459	0.295	0.065	0.121		
	Age	0.007	0.007	0.041	0.330		
Sexual	(Constant)	4.857	0.614		<0.001	0.166	<0.001
	Bedtime voiding frequency	0.655	0.154	0.204	<0.001		
	Global PSQI score	0.131	0.049	0.116	0.009		
	Hypertension	0.827	0.320	0.115	0.010		
	NQoL score	0.041	0.016	0.110	0.012		
	Age	0.020	0.011	0.089	0.058		

## Data Availability

The raw data presented in this study are available on request from the corresponding author. The data are not publicly available due to the research data governance policy of the institution that gave ethical approval to this study.

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
