# Peer review of "Predictive Values of Nocturia and Its Voiding Frequency on the Aging Males’ Symptoms"

_ijerph, 2022, doi:10.3390/ijerph191811632_

Round 1
Reviewer 1 Report
The relationship of melatonin levels to LUTS should be discussed. As melatonin may also influence the circadian rhythm of other hormones this should be touched upon in discussion as well.
The presentation of results is very complex and should be done in a more reader-friendly manner.
Author Response
As attached

Reviewer 2 Report
This study is an epidemiological study that examines the relationship between symptoms and AMS in men with nocturia. The research method is common and the content is easy to understand. It is a paper with content that can provide useful information to readers. However, I believe that some modifications are necessary.
1. Abstract section
Is "OSQI" a mistake for "PSQI"?
2. A statistical comparison of AMS and PSQI between the nocturia group and the control group without nocturia would be more persuasive.
3. Results section. 3.1 last sentence.
Table 2 did not allow me to underastand an association between sleep disturbance and nocturia severity.
4. Two terms are used in the text for hypertension: HTN and HBP. Please unify. Also, please unify "Nqol" and "NQoL".
5. As the authors also stated in the paper, the fact that testosterone levels were not measured in this study is a major limitation. Therefore, I think it would be better to show the values of AMS and NQoL in the group of patients without nocturia to compare with the results in nocturia group.
Author Response
As attached.

Reviewer 3 Report
The authors analyzed the community‐based survey on nocturia in China and its relationship to male aging in this study. The AMS scale evaluates the health-related quality of life and symptoms in aging males and a higher AMS score indicates the testosterone deficiency related to aging. The authors showed that the AMS score is higher in people with higher nighttime voiding frequency. This point of view is interesting, as the relationship between nocturia has not been evaluated. I have several suggestions to improve the manuscript.
1. In the survey, how to count the frequency of nocturia, number, or choice (1, 2, 3, >=4)? It should be described.
2. The AMS sexual score and PSQI increase from nocturia frequency 1 to 3, but it decreases in people with nocturia >=4. To find out the potential confounder, I recommend the author stratify the background characteristics based on the number of nocturia (1,2,3, and >=4) and revise table 1.
3. Also, why these two measures were lower in people with nocturia >= 4 should be explained.
4. If the frequency of nocturia is not a continuous variable, the authors should describe what dummy variable is applied to the frequency of nocturia in multiple regression analysis.
5. The method of stepwise selection (p-value, AIC, or others) should be described.
6. (Line 148) table 2 didn't describe the percentage of poor sleepers.
7. The conclusion is not straightforward. This study shows the frequency of nocturia was associated with the AMS score, which doesn't mean the AMS score increases due to nocturia. It is more appropriate to consider that both nocturia and male aging symptoms are caused by a shared cause (testosterone deficiency). Thus, "nocturia has impacted on the HRQoL of adult male", and the suggestion to include "nocturia when measuring the male‐specific HRQoL related to aging" seems overclaimed. The natural conclusion of this study is that men complaining of nocturia should also be screened for male testosterone deficiency.
Minor)
1. "Nqol" -> "NQoL"
2. Line 131 "frequency. ," -> "frequency,"
Author Response
As attached.

Round 2
Reviewer 2 Report
I think the authors have answered my question well. I have no further questions.